# A nanoluciferase SARS-CoV-2 for rapid neutralization testing and screening of anti-infective drugs for COVID-19

Xuping Xie [1,8✉], Antonio E. Muruato[1,2,8], Xianwen Zhang[1], Kumari G. Lokugamage[2], Camila R. Fontes-Garfias[1], Jing Zou[1], Jianying Liu[2], Ping Ren [3], Mini Balakrishnan[4], Tomas Cihlar[4], Chien-Te K. Tseng [2], Shinji Makino[2], Vineet D. Menachery[2,3,5], John P. Bilello [4✉] & Pei-Yong Shi [1,5,6,7✉]

A high-throughput platform would greatly facilitate coronavirus disease 2019 (COVID-19) serological testing and antiviral screening. Here we present a high-throughput nanoluciferase severe respiratory syndrome coronavirus 2 (SARS-CoV-2-Nluc) that is genetically stable and replicates similarly to the wild-type virus in cell culture. SARS-CoV-2-Nluc can be used to measure neutralizing antibody activity in patient sera within 5 hours, and it produces results in concordance with a plaque reduction neutralization test (PRNT). Additionally, using SARS-CoV-2-Nluc infection of A549 cells expressing human ACE2 receptor (A549-hACE2), we show that the assay can be used for antiviral screening. Using the optimized SARS-CoV-2-Nluc assay, we evaluate a panel of antivirals and other anti-infective drugs, and we identify nelfinavir, rupintrivir, and cobicistat as the most selective inhibitors of SARS-CoV-2-Nluc ($EC_{50}$ 0.77 to 2.74 μM). In contrast, most of the clinically approved antivirals, including tenofovir alafenamide, emtricitabine, sofosbuvir, ledipasvir, and velpatasvir were inactive at concentrations up to 10 μM. Collectively, this high-throughput platform represents a reliable tool for rapid neutralization testing and antiviral screening for SARS-CoV-2.

[1] Department of Biochemistry and Molecular Biology, University of Texas Medical Branch, Galveston, TX, USA. [2] Department of Microbiology and Immunology, University of Texas Medical Branch, Galveston, TX, USA. [3] Department of Pathology, University of Texas Medical Branch, Galveston, TX, USA. [4] Gilead Sciences, Inc., Foster City, CA, USA. [5] Institute for Human Infections and Immunity, University of Texas Medical Branch, Galveston, TX, USA. [6] Sealy Institute for Vaccine Sciences, University of Texas Medical Branch, Galveston, TX, USA. [7] Sealy Center for Structural Biology & Molecular Biophysics, University of Texas Medical Branch, Galveston, TX, USA. [8] These authors contributed equally: Xuping Xie, Antonio E. Muruato. ✉email: xuxie@UTMB.edu; john.bilello@gilead.com; peshi@UTMB.edu

Severe acute respiratory syndrome coronavirus 2 (SARS-CoV-2) emerged in Wuhan, China in late 2019[1,2] and caused global pandemic of coronavirus disease 2019 (COVID-19). Two other human coronaviruses emerged in the past two decades and caused severe respiratory syndrome, including SARS-CoV in 2002 and Middle East respiratory syndrome (MERS-CoV) in 2012[3]. In addition, four endemic human coronaviruses (i.e., OC43, 229E, NL63, and HKU1) cause common cold respiratory diseases. For COVID-19 diagnosis, nucleic acid-based RT-PCR assays have been used to identify individuals with acute viral infection. The RT-PCR assay is essential for detecting and contact tracing to control viral transmission. Given the unknown extent of asymptomatic infections, rapid and reliable serological assays are urgently needed to determine the real scale of local community infections. In addition, the ability to quickly measure neutralizing antibody levels is required to determine the immune status of previously infected individuals, to identify convalescent donors with protective antibodies for plasma therapy, and to evaluate various vaccines under development. Although various serological assay platforms have been developed [e.g., lateral flow immunoassay, ELISA, microsphere immunoassay, and vesicular stomatitis virus (VSV) pseudotyped with SARS-CoV-2 spike], the conventional plaque reduction neutralization test (PRNT) remains the gold standard of serological diagnosis because it directly measures the neutralizing antibody levels required to block an authentic viral infection. However, the low throughput and long assay turnaround time make PRNT impossible for large-scale diagnosis, representing a critical gap for COVID-19 response and countermeasure development.

The goals of this study were to (i) develop a rapid neutralization assay that maintains the gold standard of PRNT for serological COVID-19 diagnosis, (ii) establish a high-throughput assay for reliable antiviral screening, and (ii) screen exploratory and FDA-approved anti-infective drugs for potential COVID-19 repurposing. We established a nanoluciferase SARS-CoV-2 (SARS-CoV-2-Nluc) as a platform for rapid serodiagnosis and high-throughput drug screening. When used to test COVID-19 patient sera, the rapid neutralization assay yielded results commensurate with the conventional PRNT. A version of the SARS-CoV-2-Nluc infection assay has also been developed for high throughput screening of antivirals and validated using known SARS-CoV-2 inhibitors such as remdesivir and chloroquine. The developed assay was employed to test a collection of approved and investigational anti-infective drugs, including established antivirals against HIV and HCV.

## Results

**A stable SARS-CoV-2-Nluc**. Using an infectious cDNA clone of SARS-CoV-2 (strain 2019-nCoV/USA_WA1/2020)[4], we engineered nanoluciferase (Nluc) gene at the OFR7 of the viral genome (Fig. 1a). The insertion site of Nluc at ORF7 is based on our recent success of mNeonGreen reporter SARS-CoV-2[4]. Seven cDNA fragments spanning the SARS-CoV-2 genome were ligated in vitro to generate a full-genome Nluc cDNA. A T7 promoter was engineered to in vitro transcribe the full-length Nluc viral RNA. The RNA transcript was highly infectious after electroporation into Vero E6 cells (African green monkey kidney epithelial cells), producing $10^7$ plaque-forming units (PFU) per mL of virus. The infectious clone-derived SARS-CoV-2-Nluc developed plaques slightly larger than the wild-type recombinant SARS-CoV-2 (Fig. 1b). The SARS-CoV-2-Nluc and wild-type SARS-CoV-2 exhibited similar replication kinetics in Vero E6 cells (Fig. 1c), indicating that insertion of Nluc gene does not affect the viral replication in vitro.

To examine the stability of SARS-CoV-2-Nluc, we continuously cultured the virus for five passages on Vero E6 cells (1–2 days per passage). The passage 5 (P5) virus produced similar plaque morphology (Fig. 1d), replication kinetics (Fig. 1e), and luciferase profile as the P1 virus (Fig. 1f). Next, we performed RT-PCR to verify the retention of Nluc gene in the P1 and P5 viral genomes using two primers spanning the insertion junctions (nucleotides 25,068–28,099 of viral genome). The RT-PCR products derived from both P1 and P5 SARS-CoV-2-Nluc were 156-bp larger than that from the wild-type recombinant SARS-CoV-2 (Fig. 1g, lanes 1–3). The 156-bp difference is due to the substitution of ORF7 (368 bp) with Nluc gene (513 bp). Digestion of the RT-PCR products with BsrGI (located upstream of the Nluc insertion) and PacI (located at the C-terminal region of Nluc) generated distinct DNA fragments between the Nluc and wild-type viruses, whereas the P1 and P5 viruses produced identical digestion patterns (Fig. 1g, lanes 4–6). Furthermore, we confirmed the retention of Nluc reporter by sequencing the P1 and P5 RT-PCR products (Fig. 1h). Compared with the infectious clone-derived wild-type SARS-CoV-2[4], both P1 and P5 reporter viruses contained five single nucleotide mutations that led to amino acid changes in different viral proteins (Fig. 1h). These mutations may account for the slightly larger plaques of SARS-CoV-2-Nluc. No other mutations were recovered from the passaged viruses. Altogether, the results demonstrate that SARS-CoV-2-Nluc stably maintains the reporter gene after five rounds of passaging on Vero E6 cells. Besides Vero E6 cells, we also tested the stability of SARS-CoV-2-Nluc by passaging it for five rounds on A549 (a human alveolar epithelial cell line) stably expressing hACE2 (A549-hACE2; Fig. 2d). Restriction enzyme digestion of the RT-PCR products (with BsrGI and PacI) and sequencing results showed that the Nluc gene was retained after the virus had been passaged for five rounds (Supplementary Fig. 1).

**Human angiotensin-converting enzyme (hACE2) as a receptor for SARS-CoV-2**. We explored SARS-CoV-2-Nluc to study virus entry, serological diagnosis, and antiviral screening. Infection of Vero E6 cells with SARS-CoV-2-Nluc [multiplicity of infection (MOI) 1.0] produced a robust Nluc profile that peaked at 24 h post-infection (p.i.; Fig. 2a). As early as 1 h p.i., the Nluc signal was >10 fold above the background, suggesting that Nluc signals at early timepoints may be used to study virus entry. Thus, we evaluated the function of hACE2 in virus entry by pre-incubating Vero E6 cells with anti-hACE2 polyclonal antibodies for 1 h, followed by SARS-CoV-2-Nluc infection (Fig. 2b). The anti-hACE2 antibodies inhibited Nluc signal at 6 h p.i. in a dose-responsive manner (Fig. 2c). As a negative control, pre-treatment with antibodies against hDPP4 (a receptor for MERS-CoV infection) did not suppress Nluc activity (Fig. 2c), indicating the role of hACE2 in SARS-CoV-2 entry. To further evaluate these results, we compared the efficiencies of virus entry between naïve A549 and A549-hACE2 (Fig. 2d). At various MOIs, the Nluc signals (collected at 24 h p.i.) from A549-hACE2 cells were ~100-fold higher than those from the naïve A549 cells (Fig. 2e). Collectively, the results support hACE2 as a receptor for SARS-CoV-2 entry and demonstrate the utility of the SARS-CoV-2-Nluc to study virus entry.

**A rapid neutralization assay for COVID-19 diagnosis**. The robust early Nluc signals after SARS-CoV-2-Nluc infection (Fig. 2a) prompted us to develop a rapid neutralization assay. Figure 3a depicts the flowchart of SARS-CoV-2-Nluc neutralization assay in a 96-well format. After incubating serum samples with SARS-CoV-2-Nluc at 37 °C for 1 h, the virus-serum

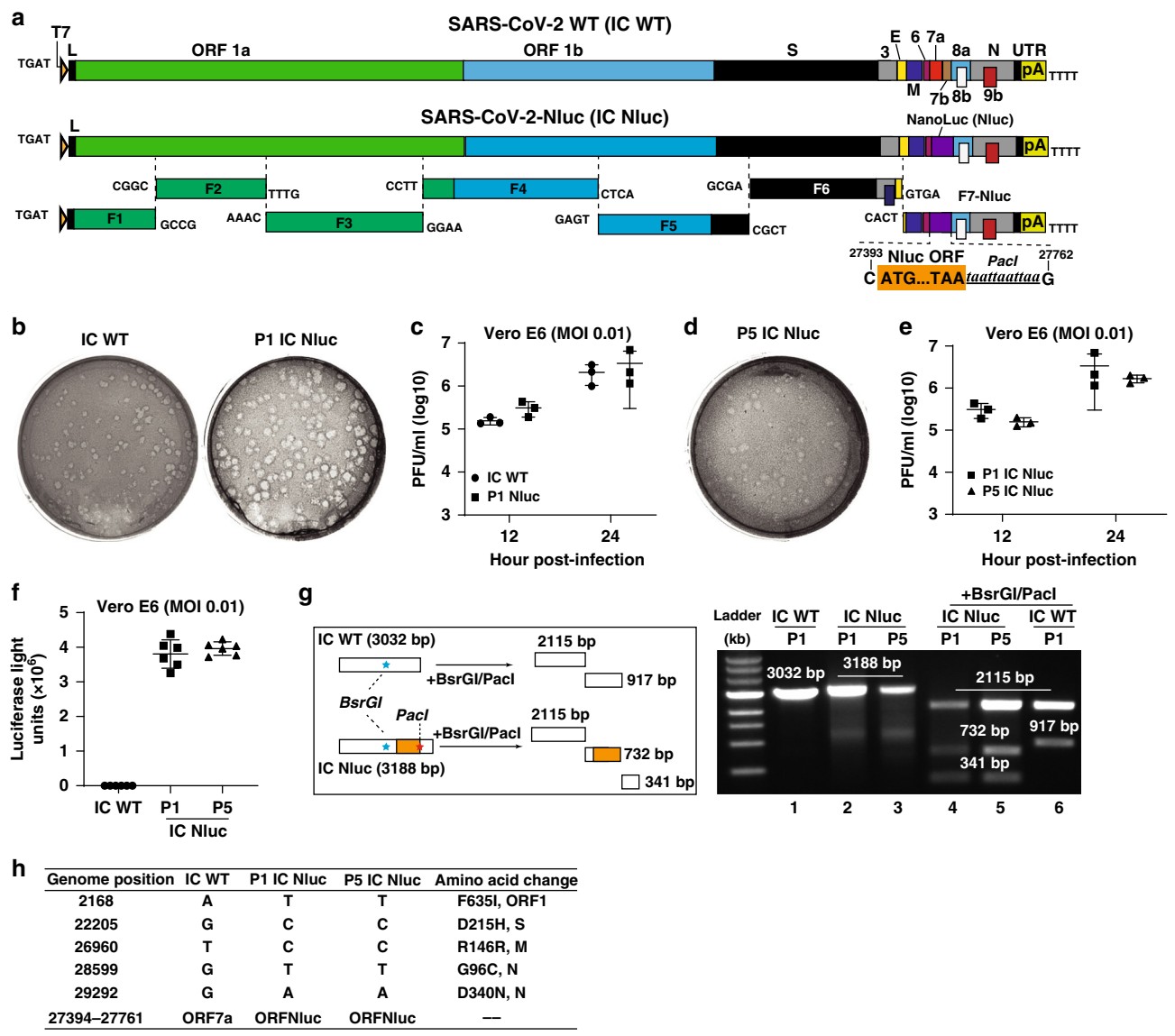

**Fig. 1 Development and characterization of SARS-CoV-2-Nluc. a** Assembly of the full-length SARS-CoV-2-Nluc cDNA. The Nanoluciferase (Nluc) gene together with a *PacI* site was placed downstream of the regulatory sequence of ORF7 to replace the ORF7 sequence. The nucleotide identities of the Nluc substitution sites are indicated. **b** Plaque morphologies of infectious clone-derived P1 SARS-CoV-2-Nluc (P1 IC Nluc) and wild-type SARS-CoV-2 (IC WT). **c** Replication kinetics. Vero E6 cells were infected with infectious clone-derived IC WT or P1 IC Nluc at MOI 0.01. Viruses in culture supernatants were quantified by plaque assay. The means ± standard deviations from three independent experiments are shown. Two-way ANOVA with correct for multiple comparisons are used for statistical analyses. **d** Plaque morphology of P5 IC Nluc. **e** Replication kinetics of P5 IC Nluc on Vero E6 cells. The means ± standard deviations from three independent experiments are shown. Two-way ANOVA with correct for multiple comparisons are used for statistical analyses. **f** Luciferase signals produced from SARS-CoV-2-Nluc-infected Vero E6 cells at 12 h post-infection. Cells were infected with viruses at MOI 0.1. The means ± standard deviations from six independent experiments are shown. One-way ordinary ANOVA test was used for statistical analyses. **g** Gel analysis of IC Nluc virus stability. The left panel depicts the theoretical results of RT-PCR followed by restriction enzyme digestion. The right panel shows the gel analysis of the RT-PCR products before (lanes 1–3) and after BsrGI/PacI digestion (lanes 4–6). **h** Summary of full-genome sequences of P1 and P5 IC Nluc viruses. Nucleotide and amino acid differences from the IC WT are indicated.

mixtures were added to Vero E6 cells (pre-seeded in a 96-well plate) at a MOI of 0.5. At 4 h p.i., Nluc signals were measured to determine the serum dilution that neutralized 50% of Nluc activity ($NT_{50}$). We chose 4 h p.i. as the assay end time because the Nluc signal at this timepoint was >100 fold above the background (Fig. 2a). The total assay time to completion was 5 h (1 h virus-serum incubation plus 4 h viral infection). Following this protocol, we tested 21 COVID-19-positive sera from RT-PCR-confirmed patients and nine COVID-19-negative human sera (collected before COVID-19 emergence; Fig. 3b). All COVID-19-positive sera (samples 1–21) showed positive $NT_{50}$ of 66–7237,

while all COVID-19-negative sera (samples 22–30) showed negative $NT_{50} < 20$, the lowest tested serum dilution. Figure 3c shows three representative neutralization curves: Nluc signals were suppressed by the positive sera in an inverse dilution-dependent manner. The results suggest that SARS-CoV-2-Nluc could be used for rapid neutralization testing.

To validate the Nluc neutralization results, we performed conventional PRNT on the same set of patient sera. The 21 COVID-19-positive samples exhibited $PRNT_{50}$ of 80–3200, and the nine COVID-19-negative samples showed $PRNT_{50} < 20$ (Fig. 3b). The neutralization results between the Nluc virus and

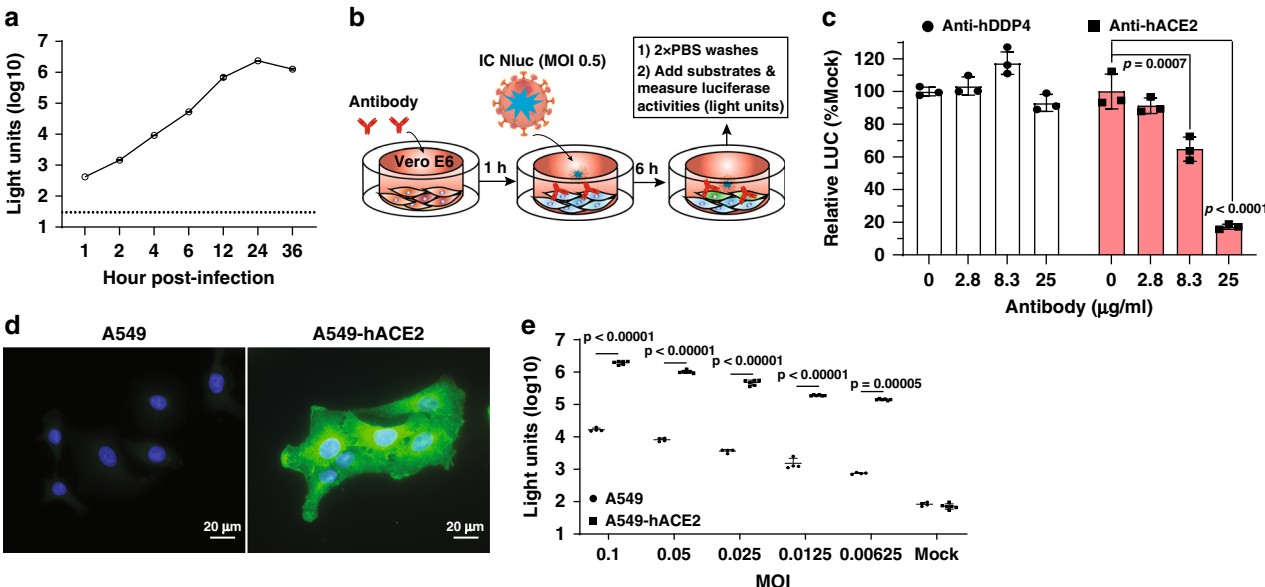

**Fig. 2 Application of SARS-CoV-2-Nluc in analyzing hACE2 as an entry receptor. a** Replication kinetics of SARS-CoV-2-Nluc (IC Nluc) on Vero E6 cells. Cells were infected with IC Nluc at MOI 1.0. At given time points, cells were harvested for luciferase signal measurement. The means and standard deviations from three independent experiments are presented. **b** Diagram to analyze hACE2 for IC Nluc entry. **c** Relative luciferase signals following infection of cells that were preincubated with anti-hDPP4 or anti-hACE2 antibodies. The luciferase signals from antibody-treated groups were normalized to those from untreated groups. The means ± standard deviations from three independent experiments are presented. One-way ANOVA was performed to analyze the statistical significance. **d** Immunofluorescence analysis of hACE2 expression in A549-hACE2 cells. At 24 h post-seeding, the cells were fixed and stained with anti-hACE2 polyclonal antibody. One representative image from six independent measurements is shown. **e** Luciferase signals from IC Nluc infected-A549 and A549-hACE2 cells. Cells were infected with indicated MOIs and luciferase signals were measured at 24 h post-infection. The means ± standard deviations from 4 to 6 independent experiments are presented. Correct for multiple comparisons using the Holm–Sidak method was performed for statistical analyses.

PRNT assays had a correlation coefficient ($R^2$) of 0.8380 (Fig. 3d). Notably, the $NT_{50}$ values from the Nluc assay are on average 3-fold higher than the $PRNT_{50}$ values form the plaque assay. Overall, the results indicate that the SARS-CoV-2-Nluc neutralization assay detects neutralizing antibodies in COVID-19 patient sera with a higher sensitivity than the conventional PRNT assay.

**A high-throughput antiviral assay for SARS-CoV-2.** Reporter viruses have been commonly used for antiviral screening[5–11]. Therefore, we developed a 96-well format antiviral assay using the SARS-CoV-2-Nluc reporter virus. Vero E6 cells were initially used in our assay development because this cell line is highly susceptible to SARS-CoV-2 infection[1]. Since COVID-19 is a respiratory disease, we also tested A549 (a human alveolar epithelial cell line) for the assay development. However, due to the low permissiveness of A549 for SARS-CoV-2-Nluc infection, we included A549-hACE2 cells to enhance viral infection in our assay (Fig. 2e). Two SARS-CoV-2 inhibitors that received the emergency use authorization in US for COVID-19 at the time of assay development, chloroquine phosphate (a malaria drug) and remdesivir (an antiviral adenosine analog prodrug)[12], were used to evaluate the assay in both Vero E6 and A549-hACE2 cells (Fig. 4). In a 3-day cytotoxicity assay, chloroquine showed $CC_{50}$ of >50 μM on both cells, whereas remdesivir had $CC_{50}$ of >50 and 32.5 μM in Vero E6 and A549-hACE2 cells, respectively (Fig. 4a, b). For testing antiviral activity, we optimized the assay conditions (12,000 Vero or A549-hACE2 cells per well and MOI 0.025) to allow for multiple rounds of viral replication in 48 h p.i. without developing significant cytopathic effect (CPE). Both chloroquine and remdesivir inhibited Nluc activity in a dose-dependent manner (Fig. 4c, d). Importantly, the $EC_{50}$ value for

remdesivir in A549-hACE2 cells (0.115 μM) was >10-fold lower than that in Vero E6 cells (1.28 μM), while the potency of chloroquine was only marginally different between the two cell lines ($EC_{50}$ 1.32 vs. 3.52 μM; Fig. 4e). This result underscores the importance of using biologically relevant cells for antiviral testing. Thus, we chose A549-hACE2 for the following high-throughput antiviral screening of additional compounds.

**Testing of clinically relevant anti-infective drugs for antiviral activity against SARS-CoV-2.** A broad selection of clinically approved and investigational antivirals and other anti-infective drugs were tested for anti-SARS-CoV-2-Nluc activities in A549-hACE2 cells. Based on their indication and/or mode of action, the tested drugs belong to four categories, including (i) antiviral nucleoside/nucleotide analogs, (ii) HIV antivirals, (iii) HCV antivirals, and (iv) other primarily anti-infective drugs.

(i) Nucleoside/nucleotide analog drugs: Eleven nucleoside analogs with antiviral activities against other viruses were evaluated for activity against SARS-CoV-2-Nluc (Table 1). Only remdesivir showed SARS-CoV-2-Nuc activity with an $EC_{50}$ and $CC_{50}$ of 0.115 and 32.7 μM, respectively, and selectivity index ($SI = CC_{50}/EC_{50}$) of 284. In comparison, the parent nucleoside of remdesivir (GS-441524) exhibited an $EC_{50}$ of 0.869 μM, a $CC_{50}$ of >50 μM, and a $SI > 57$; about 7.5-fold less potent than remdesivir. No other nucleoside analogs, including sofosbuvir or any other 2′C-methyl-substituted anti-HCV nucleosides or their prodrugs, exhibited anti-SARS-CoV-2 activity at concentrations up to 10 μM. The results agree with previous reports demonstrating potent inhibition of SARS-CoV-2 by remdesivir in physiologically relevant airway epithelial cells[13], and lack of SARS-CoV-2 inhibition by favipiravir and/or ribavirin[14–16].

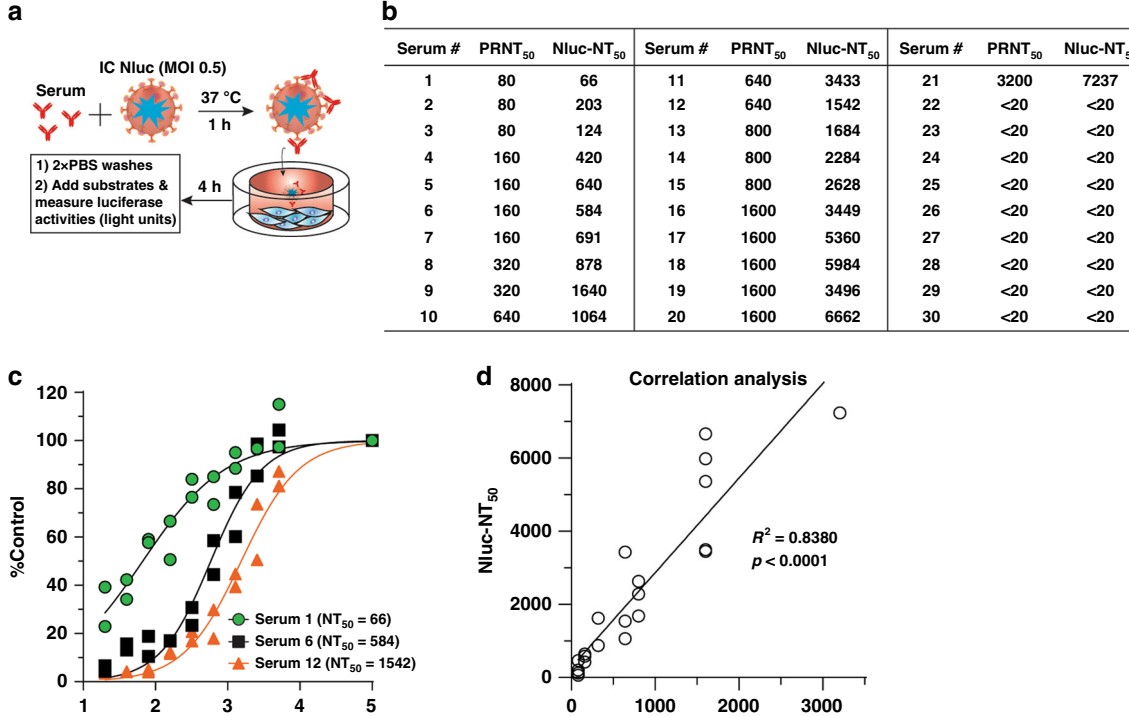

**Fig. 3 A rapid SARS-CoV-2-Nluc-based neutralization assay. a** Schematic of the rapid neutralization assay. **b** Summary of neutralizing titers as measured by PRNT and SARS-CoV-2-Nluc neutralization (Nluc-NT) assay. Serum specimens 1–21 were from COVID-19 patients with confirmed prior RT-PCR diagnosis. Serum specimens 22–30 were from non-COVID-19 individuals. **c** Representative neutralizing curves of the Nluc-NT assay. Two independent experiments are shown. The four-parameter dose–response curve was fitted using the nonlinear regression method and $NT_{50}$s were calculated in the software Prism 8. **d** Correlation analysis between the $Nluc\text{-}NT_{50}$ and $PRNT_{50}$ values. The Pearson correlation coefficient and two-tailed $p$ value from a linear regression analysis are shown.

(ii) HIV antivirals: Fifteen clinically approved antiretrovirals, including protease inhibitors (PIs), nucleoside/nucleotide reverse-transcriptase inhibitors (NRTIs), non-nucleoside reverse transcriptase inhibitors (NNRTIs), and an integrase strand-transfer inhibitor (INSTI), were assessed for their activities against SARS-CoV-2-Nluc (Table 2). Among the nine FDA-approved HIV PIs tested, nelfinavir was the only compound that inhibited SARS-CoV-2-Nluc with a sub-micromolar potency ($EC_{50}$ 0.77 µM), albeit with a relatively narrow SI of 16. Factoring in human plasma protein binding of nelfinavir[17], the projected protein adjusted potency ($paEC_{50} \sim 30$ µM) is significantly above the clinically achievable plasma concentration of the drug (Table 2). Of the remaining PIs, five were inactive (amprenavir, ritonavir, indinavir, darunavir, and atazanavir with $EC_{50} > 10$ µM) and three exhibited rather weak antiviral activity (lopinavir, saquinavir, and tipranavir with $EC_{50}$ of 8–9 µM and SI of 3–4).

Among the HIV RT inhibitors, all three NRTIs (emtricitabine, tenofovir alafenamide, and rovafovir) were inactive against SARS-CoV-2-Nluc with $EC_{50} > 10$ µM (Table 2). The two NNRTIs (rilpivirine and efavirenz) exhibited poor SI < 3.9. Bictegravir, a drug targeting HIV integrase, was inactive against SARS-CoV-2-Nluc with $EC_{50} > 10$ µM (Table 2).

(iii) HCV antivirals: Nine FDA-approved HCV drugs with diverse modes of action targeting viral protease, polymerase (both nucleotide and non-nucleoside inhibitors), or NS5A protein were tested. None of them showed any anti-SARS-CoV-2-Nluc activities with $EC_{50} > 10$ µM (Table 3).

(iv) Other classes of drugs: Ten additional clinically validated drugs, six of which are anti-infective medicines, were tested against SARS-CoV-2-Nluc (Table 4). Rupintrivir, a human rhinovirus (HRV) 3CLpro cysteine protease inhibitor, inhibited SARS-CoV-2-Nluc with $EC_{50}$ 1.87 µM, representing a 156-fold lower potency than that against HRV[18]. Niclosamide (an antihelminthic drug) showed anti-SARS-CoV-2-Nluc activity ($EC_{50}$ 0.715 µM) with low selectivity (SI 1.8). As described in Fig. 4, chloroquine exhibited selective inhibition of anti-SARS-CoV-2-Nluc ($EC_{50}$ 1.32 µM and SI > 37.9). Presatovir, a respiratory syncytial virus (RSV) fusion inhibitor, showed an $EC_{50}$ of 2.53 µM and SI of >13.5. The $EC_{50}$ of presatovir against SARS-CoV-2 is 7000-fold less potent than against RSV[19], establishing that clinical exposures are below the $EC_{50}$ determined for SARS-CoV-2[20], precluding the potential for COVID-19 therapy. Cobicistat, a selective mechanism-based inhibitor of CYP3A enzymes, weakly inhibited SARS-CoV-2-Nluc ($EC_{50}$ 2.7 µM) with a modest SI of 17.3. Oseltamivir carboxylate and baloxavir, two approved drugs targeting influenza A virus neuraminidase and endonuclease, respectively, were inactive against SARS-CoV-2-Nluc with $EC_{50} > 10$ µM. Nivocasan, an inhibitor of cellular caspases 1, 8, and 9 (treatment for hepatic fibrosis and non-alcoholic steatohepatitis related to HCV infection), as well as two inhibitors of Bruton's tyrosine kinase (BTK; treatment for lymphoma and leukemia) were also inactive against SARS-CoV-2 with $EC_{50} > 10$ µM (Table 4). Taken together, only remdesivir, chloroquine, and rupintrivir have antiviral activity against recombinant SARS-CoV-2-Nluc.

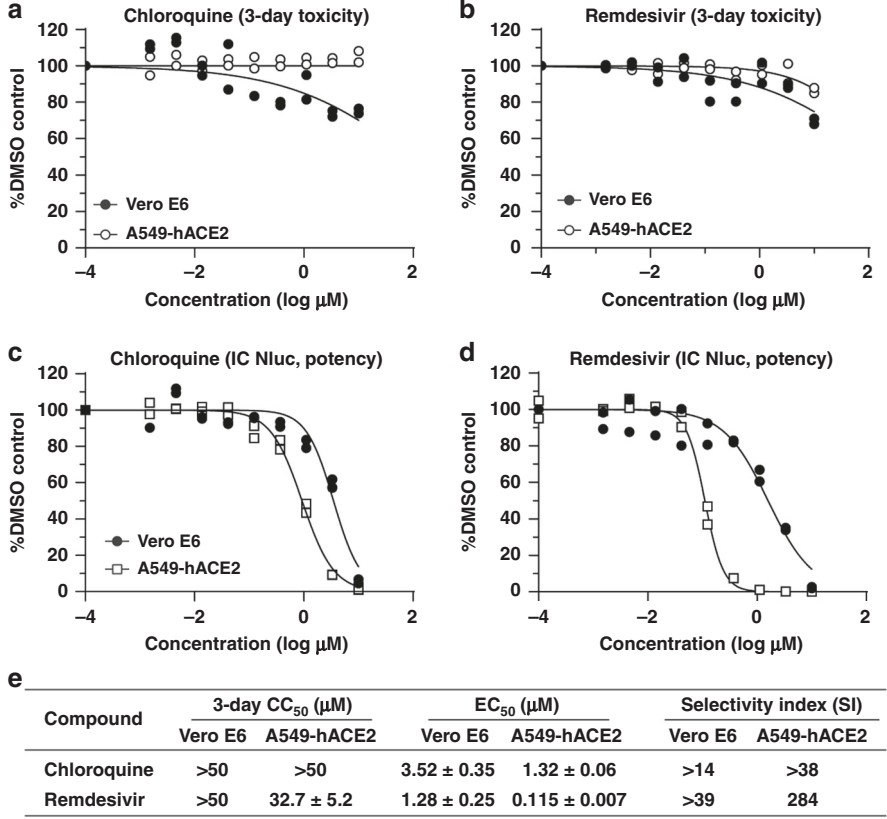

**Fig. 4 SARS-CoV-2-Nluc-based antiviral screening. a** Three-day cytotoxicity of chloroquine on Vero E6 and A549-hACE2 cells. **b** Three-day cytotoxicity of remdesivir on Vero E6 and A549-hACE2 cells. **c** $EC_{50}$ of chloroquine against SARS-CoV-2-Nluc on Vero E6 and A549-hACE2 cells. **d** $EC_{50}$ of remdesivir against SARS-CoV-2-Nluc on Vero E6 and A549-hACE2 cells. **a–d** Experiments were performed at least twice. Each time, two independent assays were performed. The plots show one of the two experiments performed in duplicates. Each replicate is shown. The four-parameter dose–response curve was fitted using the nonlinear regression method and $EC_{50}$s were calculated in the software Prism 8. **e** Summary of $CC_{50}$, $EC_{50}$, and selectivity index (SI). The mean ± standard deviations from four independent experiments are indicated. $SI = CC_{50}/EC_{50}$.

**Table 1 Nucleoside and nucleotide analogs against SARS-CoV-2-Nluc.**

| Compound name | $EC_{50}$ (μM)[a] | $CC_{50}$ (μM)[a] | SI[b] | Nucleoside/tide analog | Reference |
|---|---|---|---|---|---|
| Remdesivir (GS-5734) | 0.115 ± 0.007 | 32.7 ± 5.2 | 284 | 1′-CN-C-adenosine prodrug | 44 |
| GS-441524 | 0.869 ± 0.289 | >50 | >57 | 1′-CN-C-adenosine nucleoside | 42 |
| GS-6620 | >10 | >50 | – | 1′CN, 2′Me-C-adenosine | 45 |
| MK-0608 | >10 | >50 | – | 2′Me-7-deaza-adenosine | 35 |
| PSI-352938 | >10 | >50 | – | 2′Me-2′F-guanosine | 46 |
| Sofosbuvir | >10 | >50 | – | 2′Me, 2′F-uridine | 16 |
| ALS-8112 | >10 | >50 | – | 2′F, 4′Cl-Me-cytidine | 47 |
| Entecavir | >10 | >50 | – | Carbocyclic deoxyguanosine | 48 |
| Cidofovir | >10 | >50 | – | Acyclic cytidine phosphonate | 49 |
| Favipiravir (T-705) | >10 | >50 | – | Modified nucleobase | 50 |
| Ribavirin | >10 | >50 | – | Ribofuranosyl | – |

[a]Values are mean ± standard deviation of two independent replicate experiments in A549-hACE2 cells.
[b]Selectivity index (SI) = $CC_{50}/EC_{50}$.

## Discussion

We developed a stable reporter SARS-CoV-2-Nluc variant for rapid neutralization testing. Since neutralizing titer is a key parameter to predict immunity, the rapid SARS-CoV-2-Nluc neutralization assay will enable many aspects of COVID-19 research, including epidemiological surveillance, vaccine development, and antiviral discovery. Although the current assay was performed in a 96-well format, given the magnitude and dynamic range of Nluc signal, it can be readily adapted to a 384-well or 1536-well format for large-scale testing. Due to the amplifying nature of Nluc enzyme, the SARS-CoV-2-Nluc assay has a greater dynamic range and higher sensitivity than the fluorescent mNeonGreen virus assay[21]. Notably, when diagnosing patient sera, the SARS-CoV-2-Nluc assay generated $NT_{50}$ value on average 3-fold higher than the conventional $PRNT_{50}$. The higher sensitivity of the SARS-CoV-2-Nluc assay might be due to different endpoint readouts (plaque counts versus luminescence signal of Nluc that could accumulate in cells) or assay incubation time. Compared with the conventional PRNT assay, our reporter neutralization test has shortened the turnaround time from 3 days

**Table 2 HIV drugs against SARS-CoV-2-Nluc.**

| Inhibitor class | Compound name | EC$_{50}$ (µM)[a] | CC$_{50}$ (µM)[a] | SI[b] | Exposure (µM)[c] | Plasma protein binding (%)[d] | Reference |
|---|---|---|---|---|---|---|---|
| HIV protease (aspartyl) | Lopinavir | 9.00 ± 0.42 | 31.5 ± 2.5 | 3.5 | 15.6/8.8 | 98–99 | 51,52 |
| | Amprenavir | >10 | >50 | – | – | 90 | 53 |
| | Nelfinavir | 0.77 ± 0.32 | 12.0 ± 1.3 | 15.7 | 8.3/2.6 | >98 | 54e |
| | Ritonavir | >10 | 36.9 ± 1.7 | – | – | 98–99 | 55,56 |
| | Indinavir | >10 | >50 | – | – | 61 | 57,58 |
| | Saquinavir | 8.95 ± 0.31 | 35.1 ± 11.7 | 3.9 | 3.7/0.65 | 98 | 59e |
| | Darunavir | >10 | >50 | – | – | 95 | 60e |
| | Atazanavir | >10 | >50 | – | – | 86 | 52 |
| | Tipranavir | 8.65 ± 0.16 | 28.4 ± 0.5 | 3.3 | 130/30.8 | 99.9 | 61 |
| HIV NRTI | Emtricitabine (FTC) | >10 | >50 | – | $C_{max}$ 7.9 | 4 | 62e |
| | Tenofovir alafenamide (TAF) | >10 | >50 | – | $C_{max}$ 0.4 | 80 | 63,64 |
| | Rovafovir (GS-9131) | >10 | >50 | – | – | – | 65 |
| HIV NNRTI | Rilpivirine | 7.80 ± 1.04 | 14.6 ± 1.6 | 1.9 | 0.83/0.30 | 99.7 | 66e |
| | Efavirenz | >9.6 | 37.6 ± 10.7 | <3.9 | 12.9/5.6 | 99.5–99.8 | 67 |
| HIV integrase | Bictegravir | >10 | >50 | – | – | >99 | 68 |

[a]Values are mean ± standard deviation of two independent replicates in A549-hACE2 cells.
[b]SI = CC$_{50}$/EC$_{50}$.
[c]Values represent $C_{max}$/$C_{min}$ for human exposures in the clinic based on approved dosing schedules.
[d]Data from literature as cited.
[e]Information from product description.

**Table 3 HCV drugs against SARS-CoV-2-Nluc.**

| Inhibitor class | Compound name | EC$_{50}$ (µM)[a] | CC$_{50}$ (µM)[a] | Reference |
|---|---|---|---|---|
| HCV protease (serine) | GS-9256 | >10 | 31.8 ± 10.9 | 69 |
| | GS-9451 | >10 | >50 | 70 |
| | Voxilaprevir | >10 | 16.0 ± 1.2 | 71 |
| HCV nucleoside RdRp | Sofosbuvir | >10 | >50 | 16 |
| HCV non-nucleoside RdRp | GS-9130 | >10 | >50 | – |
| | Tegobuvir | >10 | 17.9 ± 3.1 | 72 |
| | Radalbuvir | >10 | >50 | 73 |
| HCV NS5A | Ledapisvir | >10 | >50 | 74 |
| | Velpatasvir | >10 | >50 | 75 |

[a]Values are mean ± standard deviation of two independent replicates in A549-hACE2 cells.

**Table 4 Other drug classes against SARS-CoV-2-Nluc.**

| Inhibitor class | Compound name | EC$_{50}$ (µM)[a] | CC$_{50}$ (µM)[a] | SI[b] | Reference |
|---|---|---|---|---|---|
| HRV protease (serine) | Rupintrivir | 1.87 ± 0.47 | >50 | >26.7 | 18 |
| Antihelminthic | Niclosamide | 0.715 ± 0.332 | 1.28 ± 0.23 | 1.8 | 76 |
| Antimalarial/amebicide | Chloroquine | 1.32 ± 0.36 | >50 | >37.9 | 77 |
| RSV fusion | Presatovir | 2.53 ± 0.69 | 34.0 ± 6.5 | 13.5 | 19 |
| CYP3A inhibitor | Cobicistat | 2.74 ± 0.20 | 47.3 ± 2.5 | 17.3 | 78 |
| Influenza neuraminidase | Oseltamivir carboxylate | >10 | >50 | – | 79 |
| Influenza endonuclease | Baloxavir | >10 | 47.0 ± 1.3 | – | 80 |
| Caspases 1, 8, & 9 | Nivocasan (GS-9450) | >10 | >50 | – | 81 |
| BTK | Tirabrutinib | >10 | >50 | – | 82 |
| | Ibrutinib | >10 | >50 | – | 82 |

[a]Values are mean ± standard deviation of two independent replicates in A549-hACE2 cells.
[b]SI = CC$_{50}$/EC$_{50}$.

to 5 h and increased the testing capacity. The 4-h incubation time of the Nluc assay focuses on virus entry, whereas the 3-day PRNT assay measures multiple rounds of viral replication. Despite the strengths of high throughput and speed, the current rapid neutralization assay must be performed in a biosafety level 3 (BSL-3) facility, representing a major limitation. Experiments are ongoing to attenuate SARS-CoV-2-Nluc so that the assay could be performed in a BSL-2 laboratory. Aligned with the same premise,

BSL-2 lab compatible neutralization assays have been reported using VSV pseudotyped with SARS-CoV-2 spike protein[21,22] or a medium throughput quantitative microneutralization assay based on staining of viral nucleoprotein[23].

We additionally optimized and validated the recombinant SARS-CoV-2-Nluc for high-throughput antiviral screening. Our results demonstrate that cell type could significantly affect a compound's EC$_{50}$ value, underscoring the importance of using

biologically relevant cells for drug discovery. The extent of $EC_{50}$ discrepancy from different cells was dependent on the compound's mode of action. Remdesivir $EC_{50}$ values differed by >10-fold when the assay used Vero E6 and A549-hACE2 cells. In another study, remdesivir was shown to be even more potent ($EC_{50}$ 0.01 μM) when tested on primary human airway epithelial (HAE) cells[13]. The potency differences seen between cell types are due to the differential metabolism of remdesivir in various cells. Host metabolic enzymes are required to convert the remdesivir prodrug to a monophosphate substrate, which is further metabolized by host kinases to its active triphosphate form that incorporates into viral RNA for chain termination. Vero E6 cells are less efficient in forming the active triphosphate than A549-hACE2 and primary HAE cells[13,24], leading to higher $EC_{50}$ values. The antiviral activity of chloroquine was more consistent between the two cell lines tested, indicating that its mode of action is independent of host metabolism. This highlights the need for careful and appropriate interpretation of in vitro antiviral data for compounds with different mechanisms of action such as remdesivir and chloroquine, which may appear similar in some cell types but are substantially different in cells that are more clinically relevant for SARS-CoV-2 infection.

Remdesivir has received the FDA EUA for COVID-19 treatment and is being tested in additional clinical trials, including combination therapies. In a double-blind, randomized, placebo-controlled trial involving 1063 patients hospitalized with COVID-19, patients receiving remdesivir experienced a shortened recovery time of 11 days as compared with 15 days for patients in the placebo group[25]. Besides SARS-CoV-2, remdesivir was also shown to potently inhibit SARS-CoV and MERS-CoV in cell culture and animal models[12,26–29]. For chloroquine, inconsistent results were obtained from several clinical studies with small patient numbers[30–32]. A recent retrospective multicenter study involving >1400 patients showed that treatment with hydroxychloroquine, azithromycin, or both, compared with no treatment, was not associated with significant differences in fatality rate among hospitalized patients[33]. These and other controversial results prompted recent decision by FDA to revoke the EUA for chloroquine and hydroxychloroquine (https://www.fda.gov/news-events/press-announcements/coronavirus-covid-19-update-fda-revokes-emergency-use-authorization-chloroquine-and).

Using the validated SARS-CoV-2-Nluc/A549-hACE2 infection assay, we screened a collection of clinically relevant antivirals and anti-infective drugs. In addition to remdesivir, its parent nucleoside (GS-441524), chloroquine, nelfinavir (HIV protease inhibitor), rupintrivir (HRV protease inhibitor), and cobicistat (a pharmacoenhancer and inhibitor of CYP450) were identified as the most potent and selective inhibitors among the tested compounds with $EC_{50}$ values ranging from 0.77 to 2.74 μM and SI > 15-fold. In studies with HIV in vitro, a 40-fold shift in the antiviral $EC_{50}$ was reported when assays were conducted in the presence of 50% human serum[17], an effect also likely relevant for COVID-19. Based on their antiviral potencies established in vitro, it is unlikely that nelfinavir or cobicistat would exert major clinical effects in COVID-19 patients at the current clinically approved doses, since their systemic free drug levels based on total plasma concentration and established plasma protein binding are below their measured in vitro $EC_{50}$ for SARS-CoV-2-Nluc[34,35]. Rupintrivir is a selective covalent inhibitor of HRV 3CLpro cysteine protease[18], and thus may inhibit SARS-CoV-2 through blocking the main 3CLpro cysteine protease activity. Rupintrivir has potent activity in vitro against HRV that is approximately 100-fold better compared to SARS-CoV-2[36]. It has been tested clinically as an intranasal spray for the treatment of HRV-associated common cold[37], but there is no clinical experience with either systemic or inhaled administration of rupintrivir.

Hence, further studies would be required to better understand rupintrivir's mode of action, efficacy in animal models, and potential clinical benefits in COVID-19 patients depending on the route of administration.

Several antiviral drugs approved for the treatment of HIV or HCV have been suggested to be potentially useful for the treatment of COVID-19[38,39]. These include in particular, sofosbuvir either alone[39,40] or in combination with velpatasvir[34], in addition to HIV NNRTIs tenofovir[41] and emtricitabine[38,39]. Their activities against SARS-CoV-2 were postulated primarily based on computational modeling of their interactions with the viral RdRp. Our results clearly demonstrate the lack of antiviral activity of this group of drugs against SARS-CoV-2; therefore, these drugs do not justify clinical studies in COVID-19 patients.

In summary, we have developed a stable recombinant SARS-CoV-2-Nluc for use in rapid neutralization testing and high-throughput antiviral drug discovery. Using the optimized and validated high-throughput infection assay, we screened a collection of approved and investigational antivirals and other anti-infective drugs. Among the tested agents, rupintrivir was identified as a selective in vitro inhibitor of SARS-CoV-2 that might be considered for further studies to fully establish its potential for the treatment of COVID-19.

## Methods

**Cell lines.** African green monkey kidney epithelial cells Vero E6 (ATCC®CRL-1586) and Vero (ATCC®CCL-81) were purchased from the American Type Culture Collection (ATCC, Bethesda, MD) and maintained in a high-glucose Dulbecco's modified Eagle's medium (DMEM) supplemented with 10% fetal bovine serum (FBS; HyClone Laboratories, South Logan, UT) and 1% penicillin/streptomycin (P/S; 10,000 U/mL). Human alveolar epithelial cell line (A549) was maintained in a high-glucose DMEM supplemented with 10% fetal bovine serum, 1% P/S and 1% 4-(2-hydroxyethyl)-1-piperazineethanesulfonic acid (HEPES); ThermoFisher Scientific). The A549-hACE2 cells that stably express hACE2[42] were grown in the culture medium supplemented with 10 μg/mL Blasticidin S. Cells were grown at 37 °C with 5% $CO_2$. All culture medium and antibiotics were purchased from ThermoFisher Scientific (Waltham, MA). All cell lines were tested negative for mycoplasma.

**Generation of SARS-CoV-2-Nluc.** Seven subclones (pUC57-F1, pCC1-F2, pCC1-F3, pUC57-F4, pUC57-F5, pUC57-F6, and pCC1-F7) containing the cDNA fragments of SARS-CoV-2 genome described previously[43] were used in this study. A DNA fragment containing the NanoRluciferase gene followed by a PacI restriction site (taattaattaa) was amplified by PCR with primers X87 and X88. Two other cDNA fragments containing SARS-CoV-2 genome were amplified from pCC1-F7 using PCR with two primers X109/X83, X84/X112. The three DNA fragments were assembled subclone pCC1-F7 by using NEBuilder® HiFi DNA Assembly kit, resulting in subclone pCC1-F7-Nluc. Primer sequences for the construction are listed in Supplementary Table 1. All seven subclones were validated by Sanger sequencing using primers as listed in the Supplementary Table 2. To assemble the full-length infectious cDNA clone of SARS-CoV-2-Nluc, F1, F2, F3, and F4 cDNA fragments were obtained by digesting the corresponding plasmids with enzyme BsaI. F5 and F6 fragments were obtained by digesting the plasmids with enzymes Esp3I and PvuI. F7-Nluc cDNA fragment was obtained by digesting the corresponding plasmid pCC1-F7-Nluc by Esp3I and SnaBI. All fragments after restriction enzyme digestion were separated on 0.6% agarose gels, visualized under a darkreader lightbox (Clare Chemical Research, Dolores, CO), excised, and purified using the QIAquick Gel Extraction Kit (Qiagen, Germantown, MD). In vitro ligation of seven contiguous panel of cDNA was performed to assemble the full-length cDNA. After ligation, the full-length cDNA was phenol–chloroform extracted, isopropanol precipitated, and resuspended in 10 μL nuclease-free water.

RNA transcript was in vitro synthesized by the mMESSAGE mMACHINE™ T7 Transcription Kit (ThermoFisher Scientific). A SARS-CoV-2 N gene transcript was in vitro transcribed from a DNA template using the mMESSAGE mMACHINE™ T7 Transcription Kit with a 2:1 ratio of cap analog to GTP. The N gene DNA template was prepared by PCR using primer Cov-T7-N-F and primer polyT-N-R (Supplementary Table 1). To recover the recombinant SARS-CoV-2-Nluc, 20 μg of total RNA transcripts (containing both full-length RNA and short RNAs) and 20 μg N gene transcript were mixed and added to a 4-mm cuvette containing 0.8 mL of Vero E6 cells ($8 \times 10^6$) in Ingenio® Electroporation Solution (Mirus). Single electrical pulse was given with a GenePulser apparatus (Bio-Rad) with setting of 270 V at 950 μF. After 5 min recovery at room temperature, the electroporated cells were seeded into a T-75 flask and incubated at 37 °C with 5% $CO_2$. On the next day, the culture fluid was replaced with 2% FBS DMEM medium. At 48 h post-

transfection, supernatants were harvested as P0 stock virus when severe virus-mediated cytopathic effect (CPE) occurred. One milliliter of the P0 virus was inoculated to a T-175 flask containing 80% confluence Vero E6 cells. The infected cells were incubated at 37 °C with 5% $CO_2$ for 2 days. Culture supernatants (P1) were harvested when CPE occurred. The titer of the virus stock was determined by a standard plaque assay. All SARS-CoV-2-Nluc propagation and other virus-related work were performed at the BSL-3 facility at UTMB. All personnel wore powered air purifying respirators (Breathe Easy, 3M) with Tyvek suits, aprons, booties, and double gloves.

**RNA extraction, RT-PCR, and Sanger sequencing**. 250 µL of culture fluids were mixed with three volume of TRIzol™ LS Reagent (Thermo Fisher Scientific). Viral RNAs were extracted per manufacturer's instructions. The extracted RNAs were dissolved in 30 µL nuclease-free water. 11 µL RNA samples were used for reverse transcription by using the SuperScript™ IV First-Strand Synthesis System (ThermoFisher Scientific) with random hexamer primers. Nine DNA fragments flanking the entire viral genome were amplified by PCR with specific primers. The resulting DNAs were cleaned up by the QIAquick PCR Purification Kit, and the genome sequences were determined by Sanger sequencing at GENEWIZ (South Plainfield, NJ).

**hACE2 antibody blocking assay**. 15,000 Vero E6 cells per well were seeded in a white opaque 96-well plate (Corning). On the next day, cells were washed three times with PBS to remove any residual FBS and followed by 1-h treatment with goat anti-human ACE2 antibody (R&D Systems) or anti-hDDP4 antibody (R&D Systems) (both antibodies were prepared in OptiMEM medium to the given concentrations). Afterwards, cells were infected with SARS-CoV-2-Nluc (MOI 0.5). At 6 h post-infection, cells were washed twice and followed by the addition of 50 µL Nano luciferase substrate (Promega). After 5 min of incubation at room temperature, luciferase signals were measured using a Synergy™ Neo2 microplate reader (BioTek) as per the manufacturer's instructions.

**Immunofluorescence assay**. Cells were seeded on a four-well chamber slide. At 24 h post-seeding, cells were fixed and permeabilized with 0.1% Triton X-100. After 1 h blocking with PBS + 1% FBS, cellular hACE2 was probed firstly by goat anti-human ACE2 antibody (R&D Systems). After three times of PBS washes, the cells were incubated with donkey anti-goat IgG conjugated with Alexa Fluor® 488 (ThermoFisher Scientific). Finally, the fluorescence images were acquired using the Nikon Ti2-E inverted microscope armed with a ×60 objective.

**Human sera**. The research protocol regarding the use of human serum specimens was reviewed and approved by the University of Texas Medical Branch (UTMB) Institutional Review Board. The sera were leftover from UTMB's Clinical Microbiology Diagnostics Laboratory, were anonymized and donated for research without the need for written consent in agreement with IRB protocol number 20-0070. All specimens were completely de-identified from patient information. A total of 40 de-identified convalescent sera from COVID-19 patients (confirmed with viral RT-PCR positive) were tested in this study. All human sera were heat-inactivated at 56 °C for 30 min before testing.

**SARS-CoV-2-Nluc neutralization assay**. Vero E6 cells (15,000 per well in medium containing 2% FBS) were plated into a white opaque 96-well plate (Corning). At 16 h post-seeding, 30 µL of 2-fold serial diluted human sera were mixed with 30 µL of SARS-CoV-2-Nluc (MOI 0.5) and incubated at 37 °C for 1 h. Afterwards, 50 µL of virus–sera complexes were transferred to each well of the 96-well plate. After 4 h of incubation at 37 °C 5% $CO_2$, cells were washed twice followed by the addition of Nano luciferase substrate (Promega). Luciferase signals were measured using a Synergy™ Neo2 microplate reader (BioTek) per the manufacturer's instructions. The relative luciferase signal was calculated by normalizing the luciferase signals of serum-treated groups to those of the no-serum controls. The concentration that reduces the 50% luciferase signal ($NT_{50}$) were estimated by using a four-parameter logistic regression model from the Prism 8 software (GraphPad Software Inc., San Diego, CA, USA).

**Plaque reduction neutralization test**. Approximately $1.2 \times 10^6$ Vero E6 cells were seeded to each well of six-well plates. On the following day, 100 PFU of infectious clone-derived wild-type SARS-CoV-2 was incubated with serially diluted serum (total volume of 200 µL) at 37 °C for 1 h. The virus–serum mixture was transferred to the pre-seeded Vero E6 cells in six-well plate. After incubation at 37 °C for 1 h, 2 mL of 2% high gel temperature agar (SeaKem) in DMEM with 2% FBS and 1% P/S was added to the infected cells per well. After 2-day incubation, 2 mL of neutral red (1 g/L in PBS; Sigma) was added to the agar-covered cells. After another 5-h incubation, neutral red was removed, and individual plaques were counted for $NT_{50}$ calculation. Each specimen was tested in duplicates.

**SARS-CoV-2-Nluc antiviral assay**. Vero or A549-hACE2 cells (12,000 cells per well in phenol-red free medium containing 2% FBS) were plated into a white opaque 96-well plate (Corning). On the next day, 2-fold serial dilutions of

compounds were prepared in dimethyl sulfoxide (DMSO). The compounds were further diluted 100-fold in the phenol-red free culture medium containing 2% FBS. Cell culture fluids were removed and incubated with 50 µL of diluted compound solutions and 50 µL of SARS-CoV2-Nluc viruses (MOI 0.025). At 48 h post-infection, 50 µL Nano luciferase substrates (Promega) were added to each well. Luciferase signals were measured using a Synergy™ Neo2 microplate reader. The relative luciferase signals were calculated by normalizing the luciferase signals of the compound-treated groups to that of the DMSO-treated groups (set as 100%). The relative luciferase signal ($Y$-axis) versus the $\log_{10}$ values of compound concentration ($X$-axis) was plotted in software Prism 8. The $EC_{50}$ (compound concentration for reducing 50% of luciferase signal) were calculated using a nonlinear regression model (four parameters). Two experiments were performed with technical duplicates.

**Cytotoxicity assay**. Vero or A549-hACE2 cells (5000 cells per well in phenol-red free medium containing 2% FBS) were plated into a clear flat bottom 96-well plate (Nunc). On the next day, 2-fold serial dilutions of compounds were prepared in DMSO. The compounds were further diluted 100-fold. 50 µL-diluted compound solutions were added to each well of the cell plates. At 72 h post-treatment, 4 µL of Cell Counting Kit-8 (CCK-8; Sigma-Aldrich) was added to each well. After incubation at 37 °C for 90 min, absorbance at 450 nm was measured using the Cytation5 multi-mode microplate reader (BioTek). The relative cell viability was calculated by normalizing the absorbance of the compound-treated groups to that of the DMSO-treated groups (set as 100%). The relative cell viability ($Y$-axis) versus the $\log_{10}$ values of compound concentration ($X$-axis) were plotted in software Prism 8. The $CC_{50}$ (compound concentration for reducing 50% of cell viability) were calculated using a nonlinear regression model (four parameters). Two experiments were performed with technical duplicates.

**Statistics and reproducibility**. Numeric data are presented as mean ± SD, unless specified otherwise in the figure legends. Statistical analysis was performed in the Software Prism 8. The $P$ values and statistical analysis methods are indicated in corresponding figure legends. The correlation of the Nluc-$NT_{50}$ and the $PRNT_{50}$ values from plaque neutralization assay was analyzed using a linear regression model in the software Prism 8 (GraphPad). Pearson correlation coefficient and two-tailed $P$ value are calculated using the default settings in the software Prism 8. The RT-PCR followed by restriction enzyme digestion experiment (Fig. 1g and Supplementary Fig. 1) was repeated independently at least once with similar results.

**Reporting summary**. Further information on research design is available in the Nature Research Reporting Summary linked to this article.

## Data availability

All data supporting the findings in this study are detailed in the paper. Source data are provided within this paper. The nanoluciferase SARS-CoV-2 is available from the World Reference Center for Emerging Viruses and Arboviruses (WRCEVA) at the University of Texas Medical Branch (UTMB) (https://www.utmb.edu/wrceva). Alternatively, contact the corresponding authors for the reagent. The reagent can be used for research without any constraints. If used for commercial or profit purpose, please contact UTMB's Technology Office or corresponding authors. Source data are provided with this paper.

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

## Acknowledgements

We also thank colleagues at UTMB for support and discussions. A.E.M. is supported by a Clinical and Translational Science Award NRSA (TL1) Training Core (TL1TR001440) from NIH. C.R.F.-G. is supported by the predoctoral fellowship from the McLaughlin Fellowship Endowment at UTMB. S.M. was supported by NIH grants AI114657 and AI146081. V.D.M. was supported by NIH grants U19AI100625, R00AG049092, R24AI120942, and STARs Award from the University of Texas System. P.-Y.S. was supported by NIH grants AI134907, AI145617, and UL1TR001439, and awards from the Sealy & Smith Foundation, Kleberg Foundation, John S. Dunn Foundation, Amon G. Carter Foundation, Gilson Longenbaugh Foundation, and Summerfield Robert Foundation.

## Author contributions

P.-Y.S. conceived the study. X.X., A.E.M., X.Z., K.G.L., C.R.F.-G., J.Z., J.L., M.B., and J.P.B. performed the experiments. X.X., A.E.M., T.C., V.D.M., J.P.B., and P.-Y.S. analyzed the results. P.R. prepared the serum specimens. C.-T.K.T. and S.M. provided critical reagents. X.X., M.B., T.C., J.P.B., and P.-Y.S. wrote the manuscript.

## Competing interests

X.X., V.D.M., and P.-Y.S. have filed a patent on the reverse genetic system and reporter SARS-CoV-2. The authors affiliated with Gilead Sciences, Inc. are employees of the company and own company stock. Other authors declare no competing interests.

## Additional information

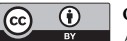

