## [Peer Review File · Nature Communications]

REVIEWER COMMENTS

Reviewer #1 (Remarks to the Author):

In their manuscript Xie and colleagues report a SARS-CoV-2 infectious clone that expresses luciferase as a reporter gene. The paper is interesting and the virus is a good tool for both serology and drug screening. However, there are some points that need the authors' attention.

Major points

1) One of the disadvantages of the test setup is, that a reader for the luciferase activity has to be present in BSL3. Many BSL3 labs won't have that. A recent paper described a medium throughput quantitative microneutralization assay based on NP staining (PMID: 32585083). These plates can be brought out of the BSL3 for staining after virus inactivation. This and the MN assay should be discussed.

2) While PRNT and the nanoluc assay results correlate, the nanoluc titers are much higher. Importantly, the PRNT actually looks at virus replication while the nanoluc assay just looks at initial entry. This should be discussed. The results from the nanoluc assays need to be taken with a grain of salt, especially when used for clinical development or development of correlates of protection.

Minor points

1) Line 24: Define 'COVID-19'

2) Line 25: Define 'SARS-CoV-2'

3) Line 57: Do we know yet that immunity is protective?

4) Line 86: Define 'PFU'

5) Throughout the manuscript: 'SARS-COV-2' is often used instead of 'SARS-CoV-2'. Please correct throughout the manuscript.

6) Line 128: Define 'MOI'

7) Line 214: 'oseltamivir'

8) Line 280: 'in vitro' should be italicized

9) Line 605: What is the actual concentration of the two antibiotics used?

10) Line 607: Define 'HEPES'

11) Line 675: Define 'DMSO'

Reviewer #2 (Remarks to the Author):

The authors have constructed a nanoluciferase (Nluc) SARS-CoV-2 for rapid detection of neutralization antibodies and antivirals against based on their previous study (PMID: 32289263). They have demonstrated that the SARS-CoV-2-Nluc stably maintains the reporter gene after five rounds of passaging on Vero E6 cells, and shown that the neutralizing activity of several sera measured using this reporter virus assay and the classic plaque reduction assay had a relatively high concordance, suggesting that this reporter virus is suitable for screening for antibodies and antivirals against SARS-CoV-2.

The manuscript is well written and the results support the conclusions presented. Nevertheless, some points listed below should be addressed.

1. The authors previously have reported a mNeonGreen reporter SARS-CoV-2 and declared that mNeonGreen reporter remains stable and could be used to study viral replication and pathogenesis as well as to develop vaccines and antiviral drugs. In this study, is the gene of mNeonGreen simply replaced with the gene of nanoluciferase? What is the difference between the two reporter viruses? The authors may compare the sensitivity and reliability of the neutralization and antiviral assays using these two different reporter viruses, and discuss the advantages and limitations of these two reporter virus assays.

2. There is a high similarity between Figure 1a in this manuscript and Figure 1A in the author's previously published article (PMID: 32289263). It is better to modify the Figure 1a accordingly.

3. The author claimed that SARS-CoV-2-Nluc stably maintains the reporter gene after five rounds of passaging on Vero E6 cells. However, Vero E6 cells are defective in type 1 IFN production. Is this reporter virus also be stable in the interferon-competent cell lines? Since the antiviral assays were performed using A549-ACE2 cells, the stability of the SARS-CoV-2-Nluc should also be detected using A549-ACE2 cells.

4. The author showed that the SARS-CoV-2-Nluc neutralization assay is rapid (5 h), but generated NT50 value on average 3-fold higher than that from the conventional PRNT. Are the difference of the NT50 values resulted from the different time period (4 h vs 48 h) of viral infection before the measurement of NT50?

5. The antiviral assay was detected at 48 h post-infection. Is this experimental condition also suitable for detection of viral entry inhibitors with function similar to that of a neutralizing antibody?

Reviewer #1

In their manuscript Xie and colleagues report a SARS-CoV-2 infectious clone that expresses luciferase as a reporter gene. The paper is interesting and the virus is a good tool for both serology and drug screening. However, there are some points that need the authors' attention.

Response: We thank the reviewer for the positive and constructive comments.

Major points

1) One of the disadvantages of the test setup is, that a reader for the luciferase activity has to be present in BSL3. Many BSL3 labs won't have that. A recent paper described a medium throughput quantitative microneutralization assay based on NP staining (PMID: 32585083). These plates can be brought out of the BSL3 for staining after virus inactivation. This and the MN assay should be discussed.

Response: We have added this MN assay and the reference (lines 246-267).

2) While PRNT and the nanoluc assay results correlate, the nanoluc titers are much higher. Importantly, the PRNT actually looks at virus replication while the nanoluc assay just looks at initial entry. This should be discussed. The results from the nanoluc assays need to be taken with a grain of salt, especially when used for clinical development or development of correlates of protection.

Response: We have added the reviewer's point to Discussion (lines 239-241).

Minor points

1) Line 24: Define 'COVID-19'

Response: Corrected (lines 24).

2) Line 25: Define 'SARS-CoV-2'

Response: Corrected (lines 25-26).

3) Line 57: Do we know yet that immunity is protective?

Response: Corrected (lines 56).

4) Line 86: Define 'PFU'

Response: Corrected (lines 85).

5) Throughout the manuscript: 'SARS-COV-2' is often used instead of 'SARS-CoV-2'. Please correct throughout the manuscript.

Response: Corrected (through spelling check).

6) Line 128: Define 'MOI'

Response: Corrected (lines 131).

7) Line 214: 'oseltamivir'

Response: Corrected (lines 217).

8) Line 280: 'in vitro' should be italicized

Response: Corrected (lines 284).

9) Line 605: What is the actual concentration of the two antibiotics used?

Response: We have added the actual antibiotics concentrations (lines 316).

10) Line 607: Define 'HEPES'

Response: Corrected (lines 317-318).

11) Line 675: Define 'DMSO'

Response: Corrected (lines 385).

Reviewer #2 (Remarks to the Author):

The authors have constructed a nanoluciferase (Nluc) SARS-CoV-2 for rapid detection of neutralization antibodies and antivirals against based on their previous study (PMID: 32289263). They have demonstrated that the SARS-CoV-2-Nluc stably maintains the reporter gene after five rounds of passaging on Vero E6 cells, and shown that the neutralizing activity of several sera measured using this reporter virus assay and the classic plaque reduction assay had a relatively high concordance, suggesting that this reporter virus is suitable for screening for antibodies and antivirals against SARS-CoV-2.

The manuscript is well written and the results support the conclusions presented. Nevertheless, some points listed below should be addressed.

Response: We thank the reviewer for the positive and constructive suggestions.

1. The authors previously have reported a mNeonGreen reporter SARS-CoV-2 and declared that mNeonGreen reporter remains stable and could be used to study viral replication and pathogenesis as well as to develop vaccines and antiviral drugs. In this study, is the gene of mNeonGreen simply replaced with the gene of nanoluciferase? What is the difference between the two reporter viruses? The authors may compare the sensitivity and reliability of the neutralization and antiviral assays using these two different reporter viruses, and discuss the advantages and limitations of these two reporter virus assays.

Response: We have addressed these questions in Discussion and Results sections. (i) The nanoluciferase is engineered at the same location as the previous mNeonGreen (lines 80-81). (ii) The difference between nanoluciferase and mNeonGreen is their distinct reporting feature (i.e., luciferase enzyme versus fluorescent signal; lines 232-234). (iii) The nanoluciferase reporter has a greater dynamic range and higher sensitivity than the mNeonGreen reporter (lines 232-

234).

2. There is a high similarity between Figure 1a in this manuscript and Figure 1A in the author's previously published article (PMID: 32289263). It is better to modify the Figure 1a accordingly.

Response: Figure 1a has been modified to differentiate from the previous diagram.

3. The author claimed that SARS-CoV-2-Nluc stably maintains the reporter gene after five rounds of passaging on Vero E6 cells. However, Vero E6 cells are defective in type 1 IFN production. Is this reporter virus also be stable in the interferon-competent cell lines? Since the antiviral assays were performed using A549-ACE2 cells, the stability of the SARS-CoV-2-Nluc should also be detected using A549-ACE2 cells.

Response: After the submission of the manuscript, we have already passaged the SARS-CoV-2-Nluc on A549-hACE2 cells for five rounds. The reporter gene was stably maintained. The results have been added to the revised manuscript as Supplementary Fig. 1 (lines 107-111).

4. The author showed that the SARS-CoV-2-Nluc neutralization assay is rapid (5 h), but generated NT50 value on average 3-fold higher than that from the conventional PRNT. Are the difference of the NT50 values resulted from the different time period (4 h vs 48 h) of viral infection before the measurement of NT50?

Response: We agree with the reviewer's comment and have added this point to Discussion (lines 237-241).

5. The antiviral assay was detected at 48 h post-infection. Is this experimental condition also suitable for detection of viral entry inhibitors with function similar to that of a neutralizing antibody?

Response: Yes. The assay incubation time of 48 h allows multiple rounds of virus infection. Thus, the assay allows to test inhibition of virus entry inhibitors. We have added this point in text (lines 126). Indeed, Fig. 2 demonstrates the assay can be used to show the inhibition of human ACE2 antibodies against SARS-CoV-2 entry.

REVIEWERS' COMMENTS

Reviewer #2 (Remarks to the Author):

The authors have satisfactorily addressed all comments of the reviewers and made the requested modifications to the manuscript, leading to significant improvement of the paper, which meets the high quality standards of Nature Communication. Therefore, I recommend immediate acceptance of the paper for publication in Nature Communication.